# Switching the spin cycloid in BiFeO₃ with an electric field

Peter Meisenheimer [1]✉, Guy Moore[1,2], Shiyu Zhou [3], Hongrui Zhang [1], Xiaoxi Huang [1], Sajid Husain [1,2], Xianzhe Chen [1,2], Lane W. Martin [1,2,4,5], Kristin A. Persson [1,6], Sinéad Griffin [2,6], Lucas Caretta [7], Paul Stevenson [8]✉ & Ramamoorthy Ramesh [1,2,4,9]

Bismuth ferrite (BiFeO₃) is a multiferroic material that exhibits both ferroelectricity and canted antiferromagnetism at room temperature, making it a unique candidate in the development of electric-field controllable magnetic devices. The magnetic moments in BiFeO₃ are arranged into a spin cycloid, resulting in unique magnetic properties which are tied to the ferroelectric order. Previous understanding of this coupling has relied on average, mesoscale measurements. Using nitrogen vacancy-based diamond magnetometry, we observe the magnetic spin cycloid structure of BiFeO₃ in real space. This structure is magnetoelectrically coupled through symmetry to the ferroelectric polarization and this relationship is maintained through electric field switching. Through a combination of in-plane and out-of-plane electrical switching, coupled with ab initio studies, we have discovered that the epitaxy from the substrate imposes a magnetoelastic anisotropy on the spin cycloid, which establishes preferred cycloid propagation directions. The energy landscape of the cycloid is shaped by both the ferroelectric degree of freedom and strain-induced anisotropy, restricting the spin spiral propagation vector to changes to specific switching events.

Multiferroic BiFeO₃ (BFO) exhibits both a large, room-temperature ferroelectric polarization and canted antiferromagnetism with deterministic coupling between these order parameters[1–6]. Thus, the magnetic nature of BFO can be controlled with electric fields, potentially paving the way for ultra-energy-efficient magnetic and spintronic devices with more favorable scaling[6–10]. Indeed, electric-field control of magnetism in coupled ferromagnetic layers using BFO been the subject of considerable attention[4,11–13], and while this functionality has long excited researchers, understanding of the process has generally relied on mesoscale imaging and transport measurements to infer the

structure and behavior of the BFO. This has left gaps in our understanding of the microscopic mechanism and details of this process.

Only recently has the opportunity to directly image and understand the spin spiral at their native scales become possible using techniques such as nitrogen-vacancy (NV) diamond-based scanning probe magnetometry (henceforth NV microscopy)[14,15]. These works have shed new light onto the nanoscopic magnetic structure of BFO[16,17]. A significant fundamental question, however, remains as to how the spin spiral evolves as the ferroelectric spontaneous polarization in BFO undergoes electric field-driven switching pathways

[1]Department of Materials Science and Engineering, University of California, Berkeley, CA, USA. [2]Materials Sciences Division, Lawrence Berkeley National Laboratory, Berkeley, CA, USA. [3]Department of Physics, Brown University, Providence, RI, USA. [4]Department of Physics and Astronomy, Department of Materials Science and Nanoengineering, Rice Advanced Materials Institute, Rice University, Houston, TX, USA. [5]Department of Chemistry, Rice University, Houston, TX, USA. [6]Molecular Foundry, Lawrence Berkeley National Laboratory, Berkeley, CA, USA. [7]School of Engineering, Brown University, Providence, RI, USA. [8]Department of Physics, Northeastern University, Boston, MA, USA. [9]Department of Physics, University of California, Berkeley, CA, USA. ✉e-mail: meisep@berkeley.edu; p.stevenson@northeastern.edu

through coupled, multi-step rotations of the polarization[11,18,19] (Fig. 1a, b, c). Here, using NV microscopy, we directly image the stray magnetic field at the surface resulting from the spin cycloid as it couples to ferroelectric domains and complex (71°, 109°, and 180°) ferroelastic and ferroelectric switching events.

In the bulk, BFO exhibits an antiferromagnetic (AFM) spin cycloid that exists in the plane defined by $P$ and the propagation direction, $k$, which points along a ⟨110⟩ that is orthogonal to $P$[20–24]. This vector connects second-nearest neighbor iron sites which, in an unperturbed G-type AFM, would be ferromagnetically coupled within a {111}[24–26]. The spin cycloid itself has been modeled as a Néel-type, rotating uncompensated magnetization, $M(r)$, that exists in the plane defined by $k$ and $P$ (i.e., the (11$\bar{2}$), where $P$ is along the [111] and $k$ along the [$\bar{1}$10], unless otherwise noted) with a period of ~65 nm (Fig. 1d). This has been described as

$$M(r) = m\left[\cos(k \cdot r)e_k + \sin(k \cdot r)e_p\right] \quad (1)$$

where $m$ is the volume-averaged magnetization, $|k| = 2\pi/\lambda$, $r$ is a coordinate in real space, and $e_k$ and $e_P$ are the unit vectors in the directions of $k$ and $P$, respectively[14]. A second-order canting also exists due to the Dzyaloshinskii-Moriya interaction (DMI) arising from the antiferrodistortive octahedral rotations[25,27–29], causing the magnetization to buckle slightly out of the $k$-$P$ plane (Fig. 1e). This second-order spin-density wave, noted here as $M_{SDW}$, can be described[16] by

$$M_{SDW}(r) = m_{DM}\cos(k \cdot r)\left(e_k \times e_p\right) \quad (2)$$

Example solutions to Eqs. 1 and 2 are provided in Supp. Figure 1. The direction of $k$, and the magnetization are thus intimately tied to the ferroelectric polarization of BFO, consistent with the fact that the spontaneous polarization is the primary order parameter of multiferroicity in this system. As the polarization is switched through

various pathways, the magnetoelectric coupling can then be evaluated in real space.

In NV microscopy, the stray magnetic fields ($B(r)$) from the sample surface perturb the energy states of a single NV center implanted into a diamond scanning probe tip. By measuring the optically detected magnetic resonance spectra of the NV center, one can detect small magnetic fields ($<2\,\mu T\sqrt{Hz}^{-1}$) with spatial resolution down to at least 10 nm. Combined with piezoresponse force microscopy (PFM), it is possible to locally map the vectorial polarization distribution, $P(r)$, in the ferroelectric domains and thereby correlate the relationship between the magnetic and ferroelectric structure in BFO. While the interaction of the cycloid with ferroelectric domains has been suggested in previous work, where poling areas of the film results in a locally uniform $k$ (the cycloid propagation direction), the interaction between ferroelectric domain walls and the cycloid has not been directly demonstrated[14]. More importantly, it is not well understood if and how the cycloid propagation direction changes during ferroelectric switching, a question especially relevant to electric-field manipulation of magnon transport[30,31] and exchange coupling across heterointerfaces[4,11]. Here, we show that direct mapping of the canted antiferromagnetic texture to the ferroelectric domains can be achieved. Of greater importance[30,31], it is shown that upon applying an electric field to switch the ferroelectric polarization, the relationship between $k$ and $P$ is conserved, with $k$ showing a strong anisotropy along the in-plane [110] and [$\bar{1}$10] which are both perpendicular to $P$. It is also observed that this anisotropy persists irrespective of the direction of electric field, through both in-plane and out-of-plane ferroelectric switching events.

## Results

### Anisotropy of the cycloid propagation

The model heterostructures studied here are ~100 nm thick, (001)-oriented BFO thin films deposited on DyScO$_3$ (DSO) (110) substrates using pulsed-laser deposition, both with and without metallic SrRuO$_3$

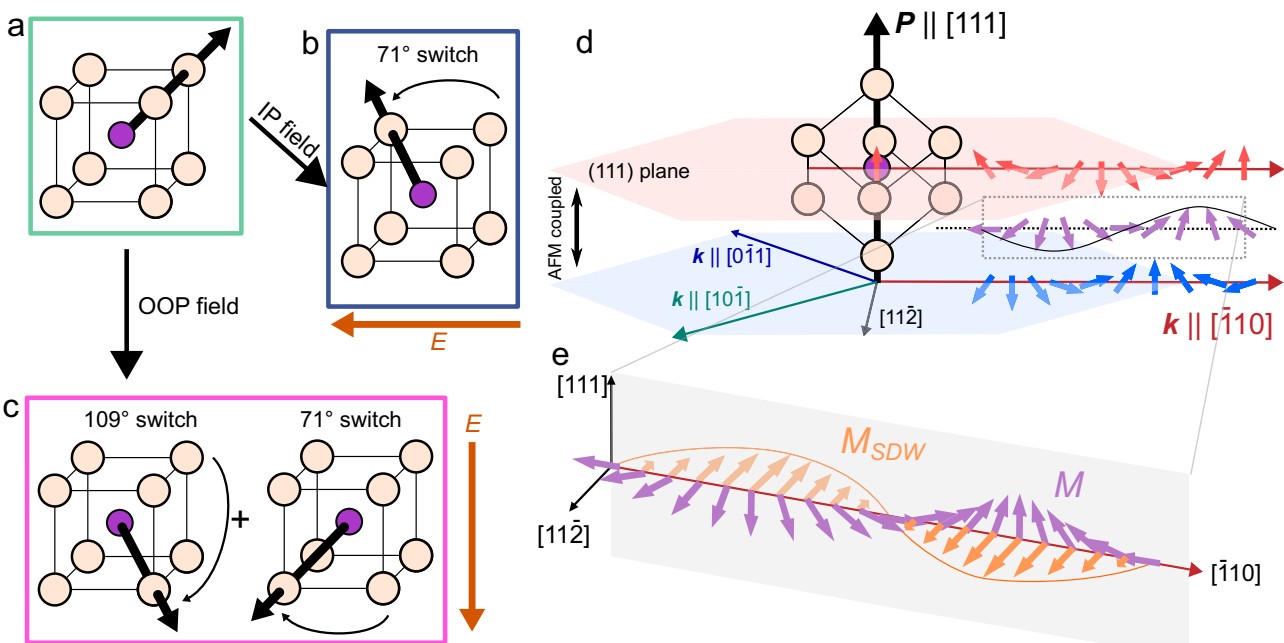

**Fig. 1 | Complex structure of BiFeO₃. a** Schematic unit cell of BFO with P along [111]. In thin films, an electric field applied in-plane, (**b**) switches the polarization by 71° to [$\bar{1}$11]. In contrast, a field applied in the out-of-plane direction, (**c**) will drive successive 71° and 109° switches, resulting in a 180° final polarization along [$\bar{1}\bar{1}\bar{1}$]. **d** Iron moments in BFO are antiferromagnetically aligned along the [111], modulated by the cycloid propagation along $k$, [$\bar{1}$10]. Other allowed directions of $k$ also lie within this (111). The canting of the AFM alignment gives rise to an uncompensated magnetization, $M(r)$, which rotates primarily in the $k$-$P$ plane with the same period as the antiferromagnetic moments, ~65 nm. **e** $M$ is further frustrated by DMI associated with the octahedral rotations, giving rise to a modulation $M_{SDW}(r)$ out of the $k$-$P$, (11$\bar{2}$) plane. The (11$\bar{2}$) plane is shown by the shaded plane and $M_{SDW}$ points along the [11$\bar{2}$] direction.

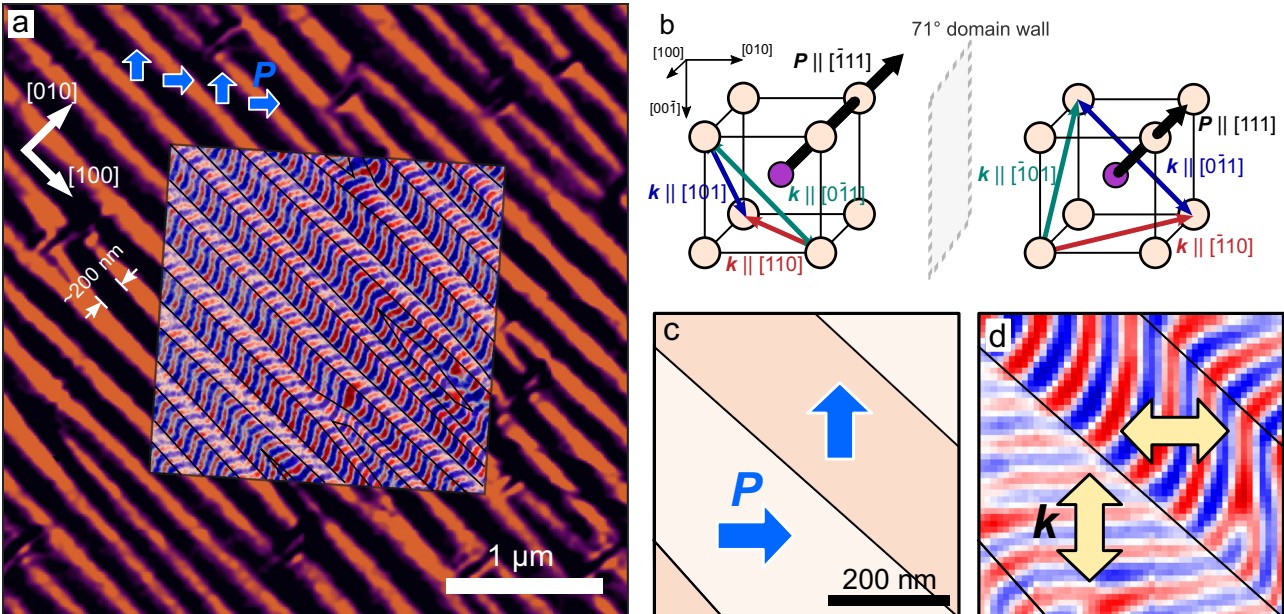

**Fig. 2 | Correlation of the magnetic cycloid. a** From PFM, BFO samples show stripe 71° ferroelectric domains along [100] that are characteristic of the material. NV images taken at the same location show that $B(\mathbf{r})$ and $P(\mathbf{r})$ can be exactly mapped where $\mathbf{k}P$. **b** Illustration showing the symmetry allowed directions of $\mathbf{k}$ and the change across the 71° ferroelectric domain wall. Because the in-plane variant of $\mathbf{k}$ (red) is favored and this is tied to the direction of $\mathbf{P}$, these domain walls in BFO give rise to a 90° rotation in $\mathbf{k}$. This is shown more closely in (**c, d**), where the local $\mathbf{P}$ can be mapped to $B(\mathbf{r})$ with continuous reorientations occurring every ~200 nm at domain walls.

bottom electrodes (Methods). In both configurations (with and without bottom electrodes), X-ray diffraction confirms that heterostructures are constrained by the substrate (Supp. Fig. 2) with a compressive strain of ~0.2% (using pseudocubic lattice constants $a_{BFO} = 3.96$, $a_{DSO} = 3.952$). Subsequent PFM imaging reveals 71° stripe-like ferroelectric domains (Fig. 2a); a commonly observed characteristic[32] of BFO. In turn, NV microscopy of the same shows the ~65 nm period sinusoidal modulation of magnetization due to the spin cycloid[14,17]. By carrying out PFM and NV microscopy at the same location, we observe that the 71° ferroelectric domain walls match exactly with changes in the cycloid propagation, switching from [110] to [$\bar{1}$10] (horizontal and vertical in Fig. 2b–d) with changes in $\mathbf{P}$. This behavior is due to the magnetoelectric coupling between $\mathbf{k}$ and $\mathbf{P}$ and demonstrates a nondegeneracy in the cycloid landscape. Generally, because $\mathbf{k}$ must remain orthogonal to $\mathbf{P}$, when $\mathbf{P}$ is along [$\bar{1}$11], $\mathbf{k}$ can propagate along [110], but when $\mathbf{P}$ changes across a domain wall to [111], $\mathbf{k}$ must reorient to an allowed direction orthogonal to the new $\mathbf{P}$, either [$\bar{1}$10], [$\bar{1}$01], or [0$\bar{1}$1] (Fig. 2b). In bulk BFO, all three of these variants are allowed[16]. Interestingly, in the thin film heterostructures studied here, $\mathbf{k}$ selects only the purely in-plane directions, i.e., [110] or [$\bar{1}$10], switching every ~200 nm corresponding to the 71° ferroelectric domains (Fig. 2c,d).

We hypothesize that the preference for the in-plane $\mathbf{k}$ vectors arises from epitaxial constraint imposed on the BFO film from the DSO substrates (e.g., differences in symmetry, lattice constant, etc.). It has been previously observed that, in thin-film BFO, the relationship between $\mathbf{k}$ and $\mathbf{P}$ can be controlled through the choice of substrate and, at small compressive strains, the cycloid follows similar behavior to bulk[14,16,23], where $\mathbf{k}$ is bound to a ⟨110⟩ that is orthogonal to $\mathbf{P}$ (so-called type-I). At higher, opposite epitaxial strains, however, a type-II cycloid can observed, which propagates along a ⟨11$\bar{2}$⟩ perpendicular to $\mathbf{P}$[16,17,33], indicating that the magnetoelastic anisotropy can follow multiple paths through phase space. It appears, then, that while constraints imposed by the DSO substrate allow for the bulk-like, type-I cycloid, it creates a strongly anisotropic landscape which inhibits out-of-plane projections of $\mathbf{k}$. This effect on the anisotropy of $\mathbf{k}$, however,

has not been determined. To better understand the impact of this anisotropy on the spin texture, density functional theory (DFT) calculations have been performed to explore the emergence of anisotropy in the spin cycloid.

While DFT simulations can provide powerful insight into the local atomic structure, predicting the ground-state magnetic order of BFO is computationally intractable using plane-wave DFT due to the long period of the cycloid (~65 nm) and the corresponding large system size required to fully simulate it[34]. A practical alternative is to use the generalized Bloch condition for $\mathbf{q}$-spirals which can account for rotations in the cycloidal plane, but this cannot predict moments that are canted out of this plane due to boundary conditions. To then help understand the ground-state magnetization of the system, here we discretize the magnetic structure into subsections manageable by first principles. Starting from a $2 \times 2 \times 2$ unit cell with G-type antiferromagnetism, where anti-aligned iron moments point along the vector $\pm L$, we systematically rotate the initial $L$ within a fixed plane (given by the angle $\phi$, Fig. 3a) to simulate the cycloid and resolve the canted magnetism. In this case, $\phi$ is in the (11$\bar{2}$), defined by $\mathbf{P}$ and $\mathbf{k}$, which hosts the cycloid. Additionally, calculations are performed applying on-site Hubbard $U$ and Hund $J$ corrections to the O-$p$ manifold, in addition to the Fe-$d$ and Bi-$p$, with further detail in Methods. We first demonstrate the validity of our approximations and methodology by reproducing the established spin texture, $L$, as reported in the literature. Specifically, we rotate $L$ within an orthogonal plane, (111) which leads to an approximately two-times greater energy cost than $L$ in (11$\bar{2}$), relative to the reference minimum energy spin quantization axis (Fig. 3b).

Initializing the iron moments along the rotation angle $\phi$ in the (11$\bar{2}$), the canted $M_{SDW}$ component of magnetization along the [11$\bar{2}$] spontaneously arises when the structure is relaxed (Supp. Fig. S3). In these simulations, $M_{SDW}$ is the net moment that comes from the canting of the atomic moments away from the initialization direction $L$ and is reported as the vector sum of the iron spins. From our simulations, $M_{SDW}$ follows the same period as the cycloid and reaches a maximum value when $L$ is parallel to [$\bar{1}$10]. This is consistent with the

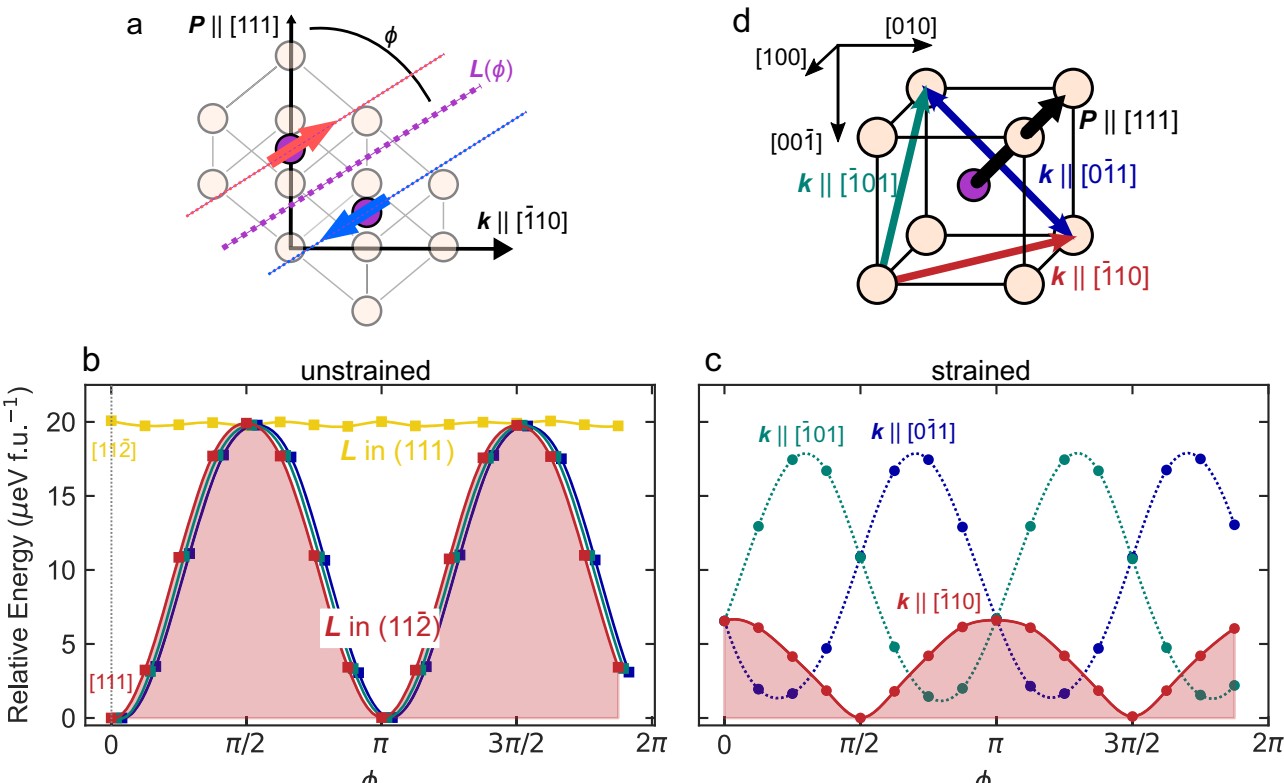

**Fig. 3 | Resolution of M$_{SDW}$ from first principles. a** Schematic showing the initialization angle $\theta$ within the cycloid plane for the antiferromagnetically aligned Fe spins. **b** Comparison of the magnetocrystalline anisotropy of the bulk unit cell when the Fe spins are rotated in the (111) and (11$\bar{2}$). This agrees with the expectation that the cycloid rotates within (11$\bar{2}$), as the mean value of the energy is 2× higher when moving the rotation to the (111) plane. **c** Relative energy along the three possible $k$ directions when the unit cell is epitaxially strained to DSO. The mean energy is 2× lower when the cycloid propagates along the in-plane [$\bar{1}$10] direction, agreeing with our experimental observation. The dotted lines show the mean energy values. **d** Schematic of the three $k$ directions in (**c**).

expectation from symmetry that $M_{SDW}$ emerges due to the DMI from octahedral rotations with their axis along the polarization direction[25,28,29], where $D_{ij} \cdot (S_i \times S_j)$ is maximized when $D_{ij}$ and $S_{i,j}$ are orthogonal, in this case $S_{i,j} \parallel k$. The value of $M_{SDW}$ reaches a maximum of 0.02 $\mu_B$, which is consistent with previous predictions[34,35] and is of the order of previous experimental results[14,16,21,28]. Discretizing the magnetic cycloid in this way then produces results that agree exactly with previous experimental and theoretical interpretations of the canted $M_{SDW}$ component of the cycloid[27-29], as well as quantitative analysis of the data presented here (Supp. Fig. S4).

With the magnetic structure reproduced using DFT, we next consider how anisotropy is introduced into the system through epitaxial constraints. Fixing the unit cell to the in-plane lattice constants of DSO, a calculation is performed where the iron moments are rotated in the planes defined by $P$ and the three possible $k$ directions (i.e., [$\bar{1}$10], [$\bar{1}$01], and [0$\bar{1}$1]). From these data, there is a clear anisotropy favoring the $k \parallel$ [$\bar{1}$10], approximately four-times lower than the original degenerate $k$. The preference for the $k \parallel$ [$\bar{1}$10] is consistent with experimental observations (Fig. 2). This is in contrast to the bulk, zero-strain state, where all possible directions of $k$ are symmetry-wise and energetically equivalent. Understanding, then, that substrate constraints intrinsically break the degeneracy of the allowed $k$ directions, one can then ask whether the state of the cycloid can be deterministically changed with an electric field and whether this anisotropy is strong enough to persist in the switched state.

**In-plane electric field switching of the cycloid**

Through application of an in-plane electric field, perpendicular to the striped 71° domain walls, we can reorient the in-plane component of polarization in BFO to produce a net polarization ($P_{net}$) along the [100]

or [$\bar{1}$00], composed of ([11$\bar{1}$] and [1$\bar{1}$$\bar{1}$]) or ([$\bar{1}$1$\bar{1}$] and [$\bar{1}$$\bar{1}$$\bar{1}$]) polarized domains, respectively. Test structures (Methods) used to apply this in-plane field and the corresponding ferroelectric switching behavior are shown in Fig. 4a,b. The ferroelectric domains measured by PFM are shown for devices, respectively poled into the $P_{net} \parallel$ [010] (Fig. 4c) and $P_{net} \parallel$ [0$\bar{1}$0] (Fig. 4d) configurations. Mapping the spin cycloid, measured through NV microscopy, to these domain images, the changes in $k$ map directly to the ferroelectric domain walls and the sense of the cycloid such that the relationship $k$ perpendicular to $P$ is preserved. Here, for example, [11$\bar{1}$] and [$\bar{1}$$\bar{1}$$\bar{1}$] polarized domains show equivalent $k$ axes along [$\bar{1}$10].

As in the case of the as-deposited films, $k$ remains parallel to the surface of the film. If the polarization of a domain reorients from [111] to [$\bar{1}$11], $k$ selects the [110] out of the three symmetry allowed axes. We believe that this is due to the biaxial strain state imposed by the DSO substrate. With this assumption, that $k$ is confined to the in-plane directions and we will observe a 90° change in $k$ under 71° ferroelectric switching, we measure the change in $k$ in situ at a single location under electric field.

With an in-plane electric field, individual ferroelastic domain walls tend to remain stationary and the polarization of individual domains reorients by 71° (Fig. 4e, Supp. Fig. S7)[4,11,30]. The stray magnetic field measured at a single location in situ and the corresponding ferroelectric domains are shown (Fig. 4f–k) where, e.g., in the center domain, the directionality of $k$ changes from [110] to [$\bar{1}$10] and back to [110] under successive switching events. Again, after ferroelectric switching, the relationship between $k$ and $P$ is conserved, with both rotating 90° about the [001]-axis. In the case of BFO, this observation is not necessarily surprising. In previous works, electrical switching of the polarization in BFO has been proposed to follow successive

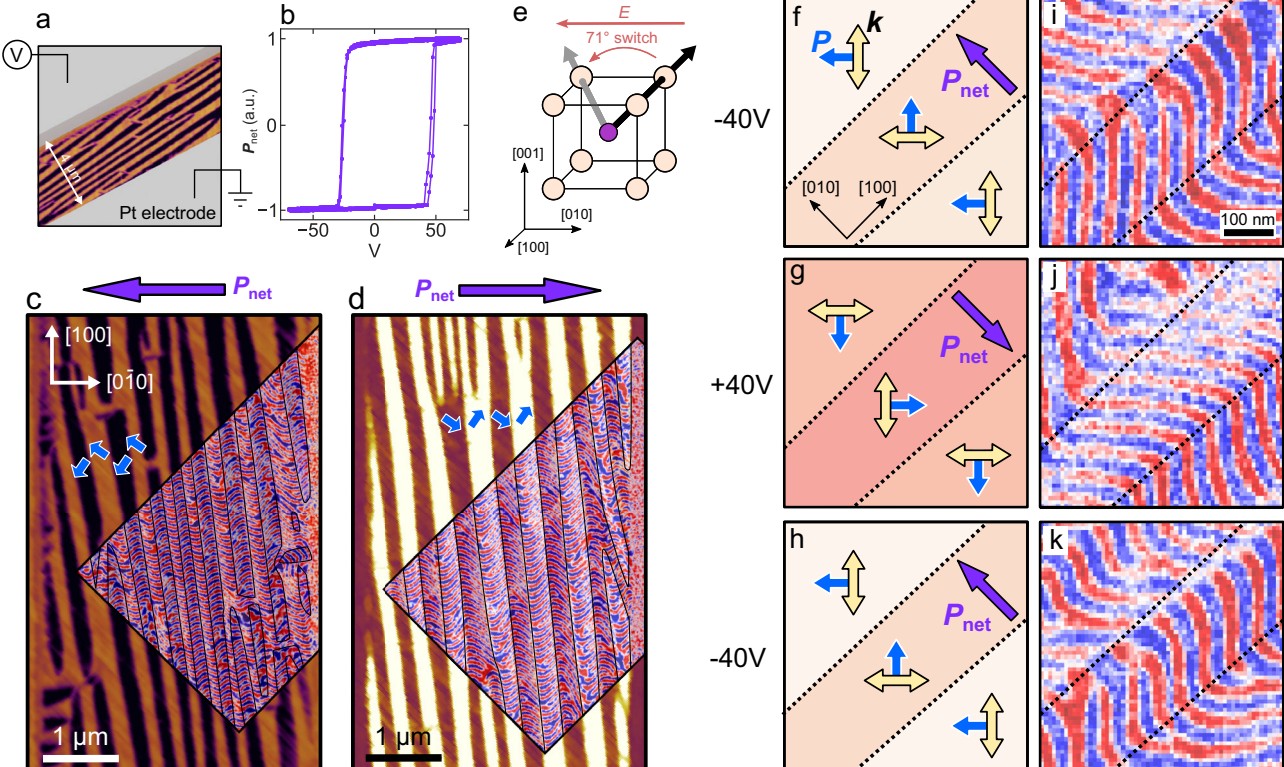

**Fig. 4 | In-plane switching of ferroelectric domains. a** Schematic of the test structure used to apply an in-plane electric field along the [010] direction, showing the orientation of the ferroelectric domain walls perpendicular to the applied field. **b** Ferroelectric hysteresis loop of the device in a, where ferroelectric switching happens at −30/+40 V. **c, d** Ferroelectric domains overlaid with NV magnetometry data for devices poled both with net polarization left and right. Ferroelectric domain walls are shown as black lines on the NV data. **e** illustration showing the 71° in-plane ferroelectric switch when the electric field is applied along [01̄0]. **f–h** Illustrations showing the 71° ferroelectric domain variants when poled along [010] (purple arrow). The in-plane projections of **P** and **k** are shown in each domain with blue and yellow arrows. **i–k** In situ NV images which map to the ferroelectric domains in (**b, c, d**). Here the 90° change in the directionality of **k**, from [110] to [1̄10] (and vice versa) can be seen clearly between switching events.

rotations of **P** (i.e., ferroelastic switching) instead of through an intermediate state which is a nonpolar configuration (i.e., ferroelectric switching)[11,18]. From this framework of the deterministic rotation of **P** and the FeO₆ octahedra, the case of an in-plane field becomes easy to understand: if **P** rotates 90° about the [001] (as in a 71° switch), and **P**⊥**k**, we would expect that **k** will also rotate correspondingly by 90° about the [001].

**Out-of-plane electric field switching of the cycloid**

In the case of an electric field applied along the [001], however, multiple switching pathways are available. Using a BFO heterostructure deposited with a SrRuO₃ back electrode, we can explore how electric fields in the out-of-plane direction, and more complex ferroelectric switching events, influence the reorientation of the spin cycloid. Examples of these complex switching pathways are illustrated in Supp. Fig. 8. Observed with X-ray diffraction, these films are commensurate to the substrate in the same fashion as the samples without SrRuO₃ (Supp. Fig. 2). An out-of-plane ([001]-oriented) electric field was applied locally to the sample using the PFM tip as the top electrode while grounding the bottom SrRuO₃ layer (Methods). A voltage of +8 V was applied to the heterostructure within a ~5 μm area, and −8 V was applied to a ~2.5 μm area within the previously switched region to return it to the same **P**net as the as-grown state (a so-called box-in-a-box structure). PFM was then used to measure the switched area at multiple scan angles to vectorize the polarization (Methods and Fig. 5a, b). Using the polarization vectors and calculating the angle between them before and after switching allows us to construct a map of ferroelectric switching events (Fig. 5c). This data can then be

compared to NV magnetometry measurements to study the dependence and anisotropy of the cycloid propagation direction on local switching events.

The map of the ferroelectric polarization direction (Fig. 5a,b), the difference between the as-deposited and switched states (Fig. 5c), and the local magnetization in both the singly (Fig. 5e–g) and doubly poled (Fig. 5h–j) regions are shown. It can be observed (Fig. 5f,i) that there are distributions of 71°, 109°, and 180° ferroelectric switching events during the poling process. As in the as-grown material, **k** is confined to the [110] and [1̄10] within the (001) and is oriented perpendicular to the final state of **P**, regardless of the local switching event being a 71°, 109°, or 180° rotation of the polarization. This is true for both the singly and doubly switched areas. Therefore, the conclusion is that **k** is determined by the final polarization vector direction, modulated by the magnetoelastic anisotropy from epitaxy.

In conclusion, we show that the antiferromagnetic spin cycloid in BFO is intimately controlled by the coupling to the spontaneous polarization, even after ferroelectric switching events. Our observations point to the key role of epitaxy (and associated magnetoelastic anisotropy) as being fundamental to the selection of specific **k** vectors of the spin cycloid. In thin films deposited on DSO, the propagation vector of the cycloid, **k**, is confined to directions orthogonal to **P**, but with no out-of-plane component, [110] and [1̄10]. This anisotropy is preserved through various ferroelectric switching events, providing a hierarchical mechanism for control of the spin cycloid. In this case, **P** is the dominant mechanism and, independently, magnetoelastic energy is a further perturbation of the structure that is tunable through lattice mismatch and film thickness.

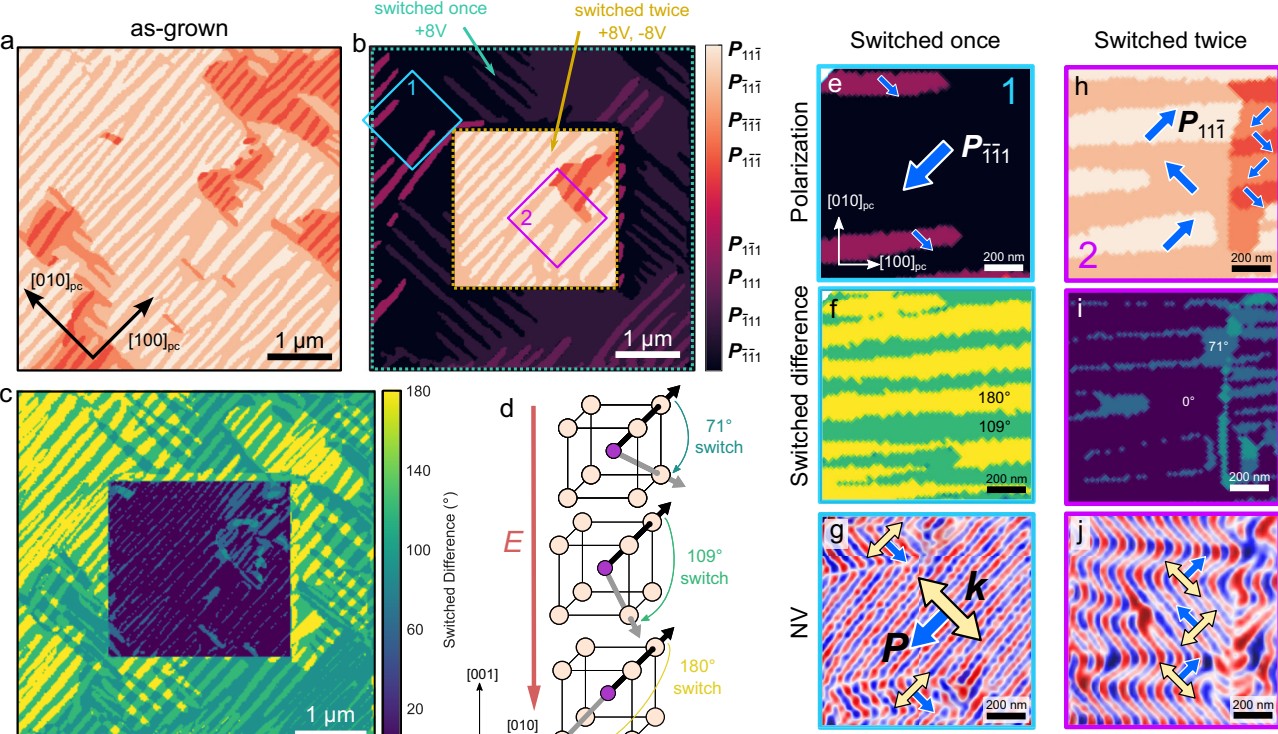

**Fig. 5 | Out-of-plane switching of the ferroelectric domains.** Vector map constructed from the PFM of BiFeO₃ samples in the (**a**) as-deposited and (**b**) out-of-plane poled configurations. The sample is poled using the PFM tip as the top electrode and ±8 V, corresponding to an electric field of ~2 MV cm⁻¹. **c** Map of the reorientation of ferroelectric domains, calculated from the difference of the polarization vectors in (**a**) and (**b**). **d** Illustration showing the different ferroelectric switching events under the application of an out-of-plane electric field. **e**, **h** zoomed-in areas of the ferroelectric polarization, shown by the blue and purple boxes labeled 1 and 2 in (**b**). The in-plane projection of the ferroelectric polarization is shown by the blue arrows. **f**, **i** Calculated switching events from c at the same locations as (**e**,**h**). **g**, **j** NV microscopy images showing the spin cycloid at the same locations in (**e**,**h**). *P* and *k* are shown by blue and yellow arrows.

## Methods

### Sample fabrication
BFO thin films were deposited on DSO substrates at 700 °C using pulsed laser deposition with a laser energy density of ~1.5 J cm⁻² in a background oxygen pressure of 90 mTorr. X-ray diffraction was measured on a Panalytical diffractometer with CuKα source. Test structures were patterned using standard lithography techniques.

Ferroelectric domains were mapped using piezoresponse force microscopy in an asylum MFP-3D, both in-plane and out-of-plane, at two orthogonal sample rotations to vectorize the polarization.

### NV microscopy
BFO samples were measured at room temperature using a commercial scanning NV microscope (Qnami ProteusQ) which combines a confocal optical microscope with a tuning-fork based atomic force microscope. Diamond tips with a parabolic taper containing single NV centers were used to increase photon collection efficiency (Quantilever MX+). The orientation of the NV center in the lab frame was determined using the method outlined in ref. 36 and a Ta(2 nm)/MgO/CoFeB(0.9 nm)/Ta(5 nm) sample with perpendicular magnetic anisotropy.

The magnetic field is quantitatively determined by measuring the optically detected magnetic resonance spectrum of the NV center at each point in space, yielding the projection of the magnetic field along the NV center axis. Large area images were collected in the qualitative "dual iso-B" mode, where the response of the NV center to two different microwave frequencies is used to track the magnetic field, as detailed in ref. 37. This dramatically reduces data acquisition time with respect to collecting the full optically detected magnetic resonance spectrum.

### Electronic structure calculation details
DFT calculations were carried out in VASP. Hubbard U and Hund J corrections for the valence states of all species were calculated using collinear DFT using a linear response (LR) workflow developed in the atomate code framework[38]. Linear response analysis was performed without the inclusion of SOC to reduce computational cost, and because SOC has been found in other systems to have a relatively small effect on the on-site corrections from LR[38]. These values were calculated to be $U, J = 5.2, 0.4$ eV for Fe-d, $U, J = 0.8, 0.8$ eV for Bi-p, and $U, J = 9.7, 1.9$ eV for O-p. Because MAE is on the order of μeV, geometry relaxation, spin-orbit coupling (SOC), and on-site Hubbard $U$ and Hund $J$ corrections are all accounted for in a holistic manner to address their interdependence. In all calculations, Hubbard and Hund corrections are applied to all outer shell manifolds (Fe-*d*, Bi-*p*, and O-*p*). These structures were calculated for the structure endpoint reported in ref. 18. Using these calculated on-site Hubbard corrections, full geometry relaxation of the structure was performed with SOC included until self-consistency was reached for an electronic energy tolerance of $10^{-6}$ eV. All calculations were performed for the $2 \times 2 \times 2$ supercell (eight times the formula unit) to accommodate the G-type antiferromagnetic structure. These computational subtleties are addressed in greater depth in the Supp. Fig. S4 and Note S1.

### Data availability
The PFM and NV data generated in this study have been deposited to Zenodo under https://doi.org/10.5281/zenodo.8310434. Other data

used in these experiments are available from the authors upon reasonable request.

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

## Acknowledgements

This work was primarily supported by the U.S. Department of Energy, Office of Science, Office of Basic Energy Sciences, Materials Sciences and Engineering Division under Contract No. DE-AC02-05-CH11231 (Codesign of Ultra-Low-Voltage Beyond CMOS Microelectronics (MicroelecLBLRamesh)) for the development of materials for low-power microelectronics. P.M., L.W.M. and R.R. additionally acknowledge funding from the Army Research Office under the ETHOS MURI via cooperative agreement W911NF-21-2-0162 and the National Science Foundation via Grant DMR-2329111. S.Z. and L.C. acknowledge funding from the Brown School of Engineering and Office of the Provost. P.S. acknowledges support from the Massachusetts Technology Collaborative, Award number #22032. G.M. acknowledges support from the Department of Energy Computational Science Graduate Fellowship (DOE CSGF) under grant DE-SC0020347. Computations in this paper were performed using resources of the National Energy Research Scientific Computing Center (NERSC), a U.S. Department of Energy Office of Science User Facility operated under contract no. DE-AC02-05CH11231. The work performed at the Molecular Foundry was supported by the Office of Science, Office of Basic Energy Sciences, of the U.S. Department of Energy under the same contract under Contract No. DEAC02-05CH11231.

## Author contributions

P.M., P.S., L.C. and R.R. designed experiments. H.Z., S.H., X.C. and X.H. synthesized and patterned samples. P.M., P.S. and S.Z. performed NV microscopy. P.M. performed PFM measurements. G.M. and S.G. performed DFT calculations. P.M., P.S., L.C., L.M., K.P. and R.R. analyzed the results and wrote the paper. All authors have contributed to manuscript revisions.

## Competing interests

The authors declare no competing interests.
