## [Peer Review File · Nature Communications]

Switching the spin cycloid in BiFeO₃ with an electric fieldEditorial Note: Parts of this Peer Review File have been redacted as indicated to remove third-party material where no permission to publish could be obtained.

REVIEWER COMMENTS

Reviewer #1 (Remarks to the Author):

In the manuscript by Meisenheimer et al., the authors reported the magnetoelastic anisotropy on the spin cycloid in BiFeO₃ films grown on DyScO₃ substrate. With the help of first principles calculations, they attributed such anisotropy to the biaxial strain imposed on the BFO film from the substrate. While the data quality is high, I am not sure if the novelty in this work can meet the standard of Nat. Commun. In Ref. 14, Gross et al. already reported the anisotropy on the spin cycloid in BiFeO₃ grown on DyScO₃ substrate, and demonstrated the effective manipulation of the cycloid propagation direction by switching the polarization orientation. Gross et al. also pointed out the compressive is the driving force of the magnetoelastic anisotropy, which has been discussed in more detailed by Sando et al. in Ref. 54.

There are some other minor issues:

1. In the caption of Fig.1, there are no explanations of panel a and b.
2. It seems that Ref. 33-43, and Ref 46-54 are not really cited in the text.

Reviewer #2 (Remarks to the Author):

Although BFO has been the most prominent material system for magnetoelectric coupling research over the last two decades, direct measurement of its nanoscopic magnetic structure has only just become possible. In this work the antiferromagnetic spin cycloid structure of BFO was observed in real space using NV microscopy. PFM and DFT calculations were employed to effectively address its coupling to the ferroelectric polarization. Scientifically this work appears to be sound and is well justified. Therefore, I think it has the potential for publication in Nature Communications. I would like the authors to clarify the following issues regarding the computational part, before I can approve its publication.

Here, the Linear response method was applied to determine the Hubbard and Hund corrections of DFT calculations. The calculated large values for O-p appear to be somewhat surprising. According to Figure S4e, it could even cause the phase inversion. Is the atomate code framework developed by the authors reliable here (may compare to other beyond-stand approaches, like HSE, GW...)? Although SOC was proved to have a small effect on other systems, is it still negligible for linear response analysis of BFO? To what extent, the choice of corrections of O-p would influence the main conclusions (i.e. substrate constraints break the degeneracy of the allowed k directions) from DFT calculations? All of these should be well addressed to prove the reasonability of the theoretical sides.

Magnetoelectric coupling should be the main topic in this work. The DFT calculations with applying electric field are thus expected to provide more dynamic details of the spin cycloid change during the polarization switching process. In this regard, the influence of electric field on the magnetocrystalline anisotropy should be further discussed at the very least.

Reviewer #3 (Remarks to the Author):

In this manuscript, authors use NV microscopy to observe the magnetic spin cycloid structure of BiFeO₃ in real space, and verified that the antiferromagnetic spin cycloid in BFO is intimately controlled by the coupling to the spontaneous polarization. This paper is of great significance to the study of magnetoelectric coupling of BiFeO₃. However, this manuscript still has some problems, and it is not suitable to publish in this stage.

1 “unless otherwise noted) with a period of ~65 nm (Figure 1a).” it should be Figure 1d.

2 “causing the magnetization to buckle slightly out of the - plane (Figure 1b).”, it should be Figure 1e.

3 “Observed with X-ray diffraction, as the samples without SrRuO₃ (Supp. Figure 1).” It should be Figure S2.

4 Please check if Figure 3b is correct, which one is (111), and which one is (11-2).

5 “At higher epitaxial strains, however, a type-II cycloid is observed, which propagates along a $\langle 11-2 \rangle$ perpendicular to P,”. However, Figure 3b is unstrained, why the mean value of the energy of (11-2) plane is low?

6 Figures S6-S8 in Supplementary were not mentioned in the manuscript.

7 In conclusion, authors state that “In thin films deposited on DSO, the propagation vector of the cycloid, k , is confined to directions orthogonal to P, but with no out-of-plane component, $[110]$ and $[1\bar{1}0]$.” $[110]$ and $[1\bar{1}0]$ belong to in-plane component.

REVIEWER COMMENTS

Responses in blue

Reviewer #1 (Remarks to the Author):

In the manuscript by Meisenheimer et al., the authors reported the magnetoelastic anisotropy on the spin cycloid in BiFeO₃ films grown on DyScO₃ substrate. With the help of first principles calculations, they attributed such anisotropy to the biaxial strain imposed on the BFO film from the substrate. While the data quality is high, I am not sure if the novelty in this work can meet the standard of Nat. Commun. In Ref. 14, Gross et al. already reported the anisotropy on the spin cycloid in BiFeO₃ grown on DyScO₃ substrate, and demonstrated the effective manipulation of the cycloid propagation direction by switching the polarization orientation. Gross et al. also pointed out the compressive is the driving force of the magnetoelastic anisotropy, which has been discussed in more detailed by Sando et al. in Ref. 54.

We thank the reviewer for their favorable opinion of our paper and comments on our measurements, however we respectfully disagree with their comment on the lack of novelty. Thus far, related works have focused almost exclusively on the relationship between magnetization and polarization in the ground state (as-deposited) of BFO, meaning that magnetolectric coupling through ferroelectric switching events remains a significant open question. This is the focus of our paper and our results clearly show that the spirals switch their axes differently in response to the ferroelectric polarization switching by 71, 109 and 180 degrees.

The primary new findings of this work are:

1. The spin structure is governed by the final polarization states, but not the polarization switching pathway, a result which has not been previously observed.
2. The ferroelectric polarization can be switched repeatedly, leading to deterministic and predictable changes in the magnetic structure.
3. The magnetic structure is further modulated by the epitaxial boundary conditions of the film, imposing a further magnetoelastic anisotropy on the spin cycloid, that leads to the selection of specific in-plane propagation directions.

Our observations point to the key role of epitaxy (and associated magnetoelastic anisotropy) as being fundamental to the selection of specific k vectors of the spin cycloid.

In previous works on the magnetic cycloid in BFO where NV magnetometry is used, the coupling between the polarization and magnetic structure is often mentioned, but not investigated with the rigor that we present here, namely investigating magnetization reorientation under successive and complex switching events.

In the seminal work of Ref. ¹, it is shown that NV magnetometry can, in fact, be used to evaluate the cycloid in BFO. The extent of electrical control shown in this work, however, is that poling the ferroelectric to a uniform polarization state results in a uniform magnetic structure where $P \perp k$. Additionally, magnetoelastic effects, which are evaluated in our manuscript, are only speculated. The relevant text from ¹ is reproduced here:

“To gain further insights into the properties of the spin cycloid in this BFO thin film, PFM was used to define a single micrometre-sized ferroelectric domain with +P1 polarization within the as-grown striped pattern (Fig. 3a), taking advantage of the trailing electric field induced by the slow scan axis of the scanning probe (Methods). The magnetic field distribution recorded above such a ferroelectric monodomain exhibits a simple periodic structure, indicating the presence of a single spin cycloid (Fig. 3b).”

In ², though NV is used, coupling to the ferroelectric order is only inferred in the ground state.

In ³ and ⁴, similar to ¹, electrical control is only shown by poling a ferroelectric domain a single time with out-of-plane fields using PFM and imaging the new state absent domain walls. In ⁴, these images are qualitatively compared to ferroelectric domain walls, but only in terms of aesthetic changes. The relevant text from ³:

“We first use PFM to draw micron-size ferroelectric domains (Supplementary Fig. 7) by virtue of the so-called trailing field^{24,25,26}. Using microdiffraction experiments, we checked that no strain difference could be detected between artificially written and as-grown striped-domains (Methods and Supplementary Fig. 8). NV magnetometry is then performed on these artificial domains to reveal the corresponding magnetic textures”

And from ⁴ is reproduced:

“Two dark $2 \times 2 \mu\text{m}^2$ squares are written by scanning the grounded conducting tip of the PFM at 45° from the [001]o direction while applying a positive dc bias (about twice the coercive bias) on the SrRuO₃ bottom electrode, which reverses the polarization to upward variants. The different directions of the slow-scan axis of the atomic force microscope enable us to control the in-plane directions of polarization thanks to the so-called trailing field [21,22], as confirmed by the two different in-plane PFM phase signals in the squares of Fig. 3(b)... From this large scanned area, we clearly distinguish the virgin zigzag pattern of the magnetic stray field inferred from the striped ferroelectric domains and areas containing only a single spin cycloid.”

While in ref ⁵, the effect of strain on the cycloid is investigated optically, this is only explained theoretically by a continuum model (which we match to our own calculations in Supp. Note S1), and our density functional theory represents an advance in calculations. Additionally, the focus

of our paper is the effect of epitaxial strain on magnetoelectric events, as the equilibrium state has been established by the above works, and these perturbations are not explored in ⁵.

In summary, though there are a number of valuable works using NV magnetometry to image the magnetic cycloid in BFO, until this study it has not been rigorously correlated to the ferroelectric order quantitatively and beyond the ground state.

[Redacted]

Figure | Previously published FE-magnetic interaction. Figures reproduced from Refs. ¹, ³, and ⁴ (a,b,c respectively), showing the approximate correlation between P and M in BFO thin films. These previous studies do not investigate the coupling as a function of repeated switching or during complex (OOP) switching events.

There are some other minor issues:

1. In the caption of Fig.1, there are no explanations of panel a and b.

We thank the reviewer for pointing this out- the Fig 1 caption has been amended.

Figure 1 | Complex structure of BiFeO₃. **a** Schematic unit cell of BFO with P along [111]. In thin films, an electric field applied in-plane, **b**, switches the polarization by 71° to $[\bar{1}\bar{1}1]$. In contrast, a field applied in the out-of-plane direction, **c**, will drive successive 71° and 109° switches, resulting in a 180° final polarization along $[\bar{1}\bar{1}\bar{1}]$. **d** Iron moments in BFO are antiferromagnetically aligned along the [111], modulated by the cycloid propagation along \mathbf{k} , $[\bar{1}10]$. Other allowed directions of \mathbf{k} also lie within this (111). The canting of the AFM alignment gives rise to an uncompensated magnetization, $M(\mathbf{r})$, which rotates primarily in the \mathbf{k} - \mathbf{P} plane with the same period as the antiferromagnetic moments, ~65 nm. **e** M is further frustrated by DMI associated with the octahedral rotations, giving rise to a modulation $M_{SDW}(\mathbf{r})$ out of the \mathbf{k} - \mathbf{P} , (11 $\bar{2}$) plane. The (11 $\bar{2}$) plane is shown by the shaded plane and M_{SDW} points along the $[11\bar{2}]$ direction.

2. It seems that Ref. 33-43, and Ref 46-54 are not really cited in the text.

These references correspond to the SI- they have been moved for clarity.

Reviewer #2 (Remarks to the Author):

Although BFO has been the most prominent material system for magnetoelectric coupling research over the last two decades, direct measurement of its nanoscopic magnetic structure has only just become possible. In this work the antiferromagnetic spin cycloid structure of BFO was observed in real space using NV microscopy. PFM and DFT calculations were employed to effectively address its coupling to the ferroelectric polarization. Scientifically this work appears to be sound and is well justified. Therefore, I think it has the potential for publication in Nature Communications. I would like the authors to clarify the following issues regarding the computational part, before I can approve its publication.

Here, the Linear response method was applied to determine the Hubbard and Hund corrections of DFT calculations. The calculated large values for O-p appear to be somewhat surprising. According to Figure S4e, it could even cause the phase inversion. Is the atomate code framework developed by the authors reliable here (may compare to other beyond-stand approaches, like HSE, GW...)? Although SOC was proved to have a small effect on other systems, is it still negligible for linear response analysis of BFO? To what extent, the choice of corrections of O-p would influence the main conclusions (i.e. substrate constraints break the degeneracy of the allowed k directions) from DFT calculations? All of these should be well addressed to prove the reasonability of the theoretical sides.

The relatively large Hubbard U value for oxygen 2p states agrees with several other studies, including Ref. ⁶. The larger magnitude of the U value for O-p compared to the transition metal d-states is explained in a recently published work by some of the current authors⁷. In this work, the expected magnitude of the Hubbard U value was reasoned by connecting the U value and the fundamental band gap of the elemental species:

“[The fundamental band gap, also known as the global chemical hardness,] is a quantity that has been tabulated many times, and using the results of Ref. ⁸, we find that for atomic oxygen its value is 11.2 eV, compared to that of the transition metal atoms, where it ranges from 5.8 eV (Ti & Zr) to 8.0 eV (Mn) if we exclude the often problematic zinc group, where it reaches 11.6 eV. This mirrors and explains the observed relatively large first-principles Hubbard U value for oxygen 2p states predicted in this and several previous studies.”

Specifically, the fundamental band gap of iron is 7.2 eV from Ref. ⁸. Therefore, the difference between fundamental band gaps is $v_{O-p} - v_{Fe-d} = 11.2 - 7.2 \text{ eV} = 4.0 \text{ eV}$, where the difference

between effective U values that we found using linear response is significantly less, $U_{\text{eff}}(\text{O-}p) - U_{\text{eff}}(\text{Fe-}d) = (9.7 - 1.9) \text{ eV} - (5.2 - 0.4) \text{ eV} = 3.0 \text{ eV}$. Therefore, using this reasoning, the absolute difference is actually less than expected (though of course, caution should be taken in making a direct comparison of these energies).

We thank the Reviewer for pointing out that a study of the role of spin-orbit coupling and linear response in Ref. ⁷ for another well known multiferroic, LiNiPO₄. Linear response calculation of U/J values for noncollinear DFT can have an effect on calculated U & J values. However, for LiNiPO₄, it only changed the U value by an amount within the respective uncertainty bars. Therefore, with this prior work, in addition to our own Ref. ⁷, we are confident that our approach is sufficient for treating the self-interaction correction error, while also balancing computational cost.

We agree with the Reviewer that the (previously unreported) effect that oxygen Hubbard parameters have on the sign of the effective anisotropy constant is interesting. This is not altogether surprising if one considers how Coulomb corrections to the local potential would shift upon the inclusion of significant O-*p* Hubbard U values in an octahedral environment. Since the local magnetic field due to spin-orbit coupling is proportional to the cross product between the momentum operator with the local electric field, corrections to the O-*p* manifold should have a corresponding effect on the anisotropy⁹. Lastly, the sign of the anisotropy constant with O-*p* Hubbard U values agrees more closely with experimental results than prior DFT calculations that did not include a correction on O-*p*, providing further justification of our approach ^{10,11}.

We next address the Reviewer's question about the effect of the O-*p* Hubbard U value on the preferred cycloid **k**-vector under epitaxial strain. We claim that O-*p* Hubbard U values will not affect this conclusion. To justify this, we turn the Reviewer's attention to Figure S5e in the SI. From Figure 3c of the main text, the inclusion of strain changes the minimum energy spin orientation, to be more consistent with the minimum found without the O-*p* Hubbard U value, plotted in Figure S4e. This can be justified based on the sign change of K_{eff} , which is dwarfed by U_{MS} under epitaxial strain, where U_{MS} is also strongly negative under compression in the (001) plane. Combining these observations, neglecting a O-*p* Hubbard U value would still stabilize **k** ~ [-110] under compressive epitaxial strain.

We note that the other 'beyond standard' DFT approaches that the Reviewer cites do not improve the description of the physics being targeted in this study (magnetic properties). GW is employed to calculate the electronic self-energy which is needed for the accurate description of electronic excitations (which are not relevant to the current work). While HSE (or more generally hybrid functionals) are far less expensive than full GW calculations, they are typically used for improving calculations of bandgaps (again, not relevant to our current work). We therefore selected an appropriate methodology for describing the physics of our system that includes well established cost-effective alternatives to describing correlated electron systems, such as linear-response U.

Magnetoelectric coupling should be the main topic in this work. The DFT calculations with applying electric field are thus expected to provide more dynamic details of the spin cycloid change during the polarization switching process. In this regard, the influence of electric field on the magnetocrystalline anisotropy should be further discussed at the very least.

Including an electric field in such noncollinear DFT calculations would indeed be an ideal study for fully determining the magnetoelectric coupling and the response to the cycloid under an explicitly applied electric field. However, because of the limitations of plane-wave DFT, doing this rigorously would be at the limits of what is possible to calculate using DFT, and beyond the scope of this work. Firstly, to describe the non-collinear magnetic order, a 40-atom unit cell with spin-orbit coupling is required. To include an electric field in plane-wave DFT calculations, one must perform a “slab” calculation in order to impose a potential gradient under periodic boundary conditions (i.e. to include a vacuum region where the potential gradient terminates), as described in the VASP documentation¹². The slab size then needs to be large enough so that the bulk properties are maintained; this includes both converging the slab size and ensuring that the bulk properties are not modified by spurious surface effects. Additionally, breaking symmetry of the bulk phase by introducing a surface in a system with antisymmetric exchange can also leave uncompensated DMI interactions, which will fundamentally change the magnetic ground-state¹³. Therefore, we would expect at a minimum, that a slab geometry of several hundreds of atoms in addition to a vacuum would be required for accurate predictions, which is computationally infeasible, especially with spin-orbit coupling.

Reviewer #3 (Remarks to the Author):

In this manuscript, authors use NV microscopy to observe the magnetic spin cycloid structure of BiFeO₃ in real space, and verified that the antiferromagnetic spin cycloid in BFO is intimately controlled by the coupling to the spontaneous polarization. This paper is of great significance to the study of magnetoelectric coupling of BiFeO₃. However, this manuscript still has some problems, and it is not suitable to publish in this stage.

We thank the reviewer for their favorable opinion of our manuscript and for pointing out the following mistakes. These have been corrected in the main text.

1. “unless otherwise noted) with a period of ~65 nm (Figure 1a).” it should be Figure 1d.

This has been corrected

2 “causing the magnetization to buckle slightly out of the - plane (Figure 1b).”, it should be Figure 1e.

This has been corrected

3 “Observed with X-ray diffraction, as the samples without SrRuO₃ (Supp. Figure 1).” It should be Figure S2.

This has been corrected

4 Please check if Figure 3b is correct, which one is (111), and which one is (11-2).

The indices in Figure 3b are correct. From both neutron and theoretical experiments^{14–18}, the cycloid points along $[\bar{1}10]$ orthogonal to $[111]$ with the spins, and thus L within the $k - P$, $(11\bar{2})$ plane. To demonstrate that our DFT calculations can generate the correct cycloid, we compare the energies when L is rotated in $(11\bar{2})$ (the established plane) and (111) (the plane normal to P). We find that the energies of the first case, L in $(11\bar{2})$, are much lower, agreeing with previous experimental reports.

Figure 3b | Comparison of the magnetocrystalline anisotropy of the bulk unit cell when the Fe spins are rotated in the (111) and $(11\bar{2})$. This agrees with the expectation that the cycloid rotates within $(11\bar{2})$, as the mean value of the energy is 2x higher when moving the rotation to the (111) plane.

5 “At higher epitaxial strains, however, a type-II cycloid is observed, which propagates along a $\langle 11\bar{2} \rangle$ perpendicular to P ,”. However, Figure 3b is unstrained, why the mean value of the energy of $(11\bar{2})$ plane is low?

In ref³, it was observed by the authors that in tensile-strained BFO on substrates such as GdScO₃, the cycloid can rotate from a $[\bar{1}10]$ propagation direction between nearest neighbor Fe sites, to a $[11\bar{2}]$ propagation direction between second nearest neighbor Fe sites. Presumably, this effect is also due to the magnetoelastic contribution, however the sign of the strain is opposite (tensile on GSO vs compressive on DSO), and thus the energy landscape will behave very differently. This statement about the type II cycloid has been modified in the text to be clearer.

It has been previously observed that, in thin-film BFO, the relationship between \mathbf{k} and \mathbf{P} can be controlled through the choice of substrate and, at small compressive strains, the cycloid follows similar behavior to bulk^{14,16,23}, where \mathbf{k} is bound to a $\langle 110 \rangle$ that is orthogonal to \mathbf{P} (so-called type-I). At higher, **opposite** epitaxial strains, however, a type-II cycloid can be observed, which propagates along a $\langle 11\bar{2} \rangle$ perpendicular to \mathbf{P} ^{16,17,33}, indicating that the magnetoelastic anisotropy can follow multiple paths through phase space.

6 Figures S6-S8 in Supplementary were not mentioned in the manuscript.

We thank the reviewer for pointing this out. These figures were referenced in an earlier version of the paper, but the references were removed during revision. These figures have been referenced on pages 8:

produces results that agree exactly with previous experimental and theoretical interpretations of the canted M_{SDW} component of the cycloid^{27–29}, as well as quantitative analysis of the data presented here (**Supp. Figure S4**).

9:

With an in-plane electric field, individual ferroelastic domain walls tend to remain stationary and the polarization of individual domains reorients by 71° (**Figure 4e, Supp. Figure S7**)^{4,11,30}.

And 10:

In the case of an electric field applied along the [001], however, multiple switching pathways are available. Using a BFO heterostructure deposited with a SrRuO₃ back electrode, we can explore how electric fields in the out-of-plane direction, and more complex ferroelectric switching events, influence the reorientation of the spin cycloid. **Examples of these complex switching pathways are illustrated in Supp. Figure 8.**

7 In conclusion, authors state that “In thin films deposited on DSO, the propagation vector of the cycloid, k , is confined to directions orthogonal to P , but with no out-of-plane component, [110] and $[1\bar{1}0]$.” [110] and $[1\bar{1}0]$ belong to in-plane component.

We are not completely sure what the reviewer is referencing here. Because of the magnetoelastic energy, the k directions along [101] and [011] are disfavored with respect to [110]. We refer to these directions as “OOP component” for [101] and [011], which have a [00L] component, and “fully IP” for [110], with only [H00] and [0K0] vectors.

References

1. Gross, I. *et al.* Real-space imaging of non-collinear antiferromagnetic order with a single-spin magnetometer. *Nature* **549**, 252 (2017).
2. Chauleau, J.-Y. *et al.* Electric and antiferromagnetic chiral textures at multiferroic domain walls. *Nat. Mater.* **19**, 386–390 (2020).
3. Haykal, A. *et al.* Antiferromagnetic textures in BiFeO₃ controlled by strain and electric field. *Nat Commun* **11**, 1704 (2020).
4. Zhong, H. *et al.* Quantitative Imaging of Exotic Antiferromagnetic Spin Cycloids in BiFeO₃ Thin Films. *Phys. Rev. Appl.* **17**, 044051 (2022).
5. Sando, D. *et al.* Crafting the magnonic and spintronic response of BiFeO₃ films by epitaxial strain. *Nature Materials* **12**, 641–646 (2013).
6. Linscott, E. B., Cole, D. J., Payne, M. C. & O’Regan, D. D. Role of spin in the calculation of Hubbard U and Hund’s J parameters from first principles. *Phys. Rev. B* **98**, 235157 (2018).
7. Moore, G. C. *et al.* High-throughput determination of Hubbard U and Hund J values for transition metal oxides via the linear response formalism. *Phys. Rev. Mater.* **8**, 014409 (2024).

8. Dong, X., Oganov, A. R., Cui, H., Zhou, X.-F. & Wang, H.-T. Electronegativity and chemical hardness of elements under pressure. *Proceedings of the National Academy of Sciences* **119**, e2117416119 (2022).
9. Sakurai, J. J. & Napolitano, J. Modern Quantum Mechanics. *Higher Education from Cambridge University Press* <https://www.cambridge.org/highereducation/books/modern-quantum-mechanics/DF43277E8AEDF83CC12EA62887C277DC> (2020) doi:10.1017/9781108587280.
10. Ramazanoglu, M. *et al.* Temperature-dependent properties of the magnetic order in single-crystal BiFeO_3 . *Phys. Rev. B* **83**, 174434 (2011).
11. Matsuda, M. *et al.* Magnetic Dispersion and Anisotropy in Multiferroic BiFeO_3 . *Phys. Rev. Lett.* **109**, 067205 (2012).
12. EFIELD - VASP Wiki. <https://www.vasp.at/wiki/index.php/EFIELD>.
13. Pylypovskiy, O. V. *et al.* Surface-symmetry-driven Dzyaloshinskii-Moriya interaction and canted ferrimagnetism in collinear magnetoelectric antiferromagnet Cr_2O_3 . Preprint at <https://doi.org/10.48550/arXiv.2310.13438> (2023).
14. Mikuszeit, N., Meckler, S., Wiesendanger, R. & Miranda, R. Magnetostatics and the rotational sense of cycloidal spin spirals. *Phys. Rev. B* **84**, 054404 (2011).
15. Lebeugle, D. *et al.* Electric-Field-Induced Spin Flop in BiFeO_3 Single Crystals at Room Temperature. *Phys. Rev. Lett.* **100**, 227602 (2008).
16. Sosnowska, I. & Zvezdin, A. K. Origin of the long period magnetic ordering in BiFeO_3 . *Journal of Magnetism and Magnetic Materials* **140–144**, 167–168 (1995).
17. Ke, X. *et al.* Magnetic structure of epitaxial multiferroic BiFeO_3 films with engineered ferroelectric domains. *Phys. Rev. B* **82**, 134448 (2010).
18. Sosnowska, I., Neumaier, T. P. & Steichele, E. Spiral magnetic ordering in bismuth ferrite. *J. Phys. C: Solid State Phys.* **15**, 4835 (1982).

REVIEWERS' COMMENTS

Reviewer #1 (Remarks to the Author):

In the rebuttal and revised manuscript, the authors have addressed my concerns and questions properly. I would recommend publication in Nat. Commun.

Reviewer #2 (Remarks to the Author):

The authors' response is convincing even though it does not fully address the issues I raised in the initial report. Considering the improvements to the paper after their revision, I can now recommend its publication in Nature Communications.

Reviewer #3 (Remarks to the Author):

The author answered my comments very well and this manuscript can be accepted

REVIEWERS' COMMENTS

Reviewer #1 (Remarks to the Author):

In the rebuttal and revised manuscript, the authors have addressed my concerns and questions properly. I would recommend publication in Nat. Commun.

We thank the reviewer for their favorable reading of our manuscript

Reviewer #2 (Remarks to the Author):

The authors' response is convincing even though it does not fully address the issues I raised in the initial report. Considering the improvements to the paper after their revision, I can now recommend its publication in Nature Communications.

We thank the reviewer for their favorable reading of our manuscript

Reviewer #3 (Remarks to the Author):

The author answered my comments very well and this manuscript can be accepted

We thank the reviewer for their favorable reading of our manuscript